# Simulation of Carbon Sink of Arbor Forest Vegetation in Henan Province of China Based on CO2FIX Model

**Kaili Cheng** [1,2,3,4], **Jing Wu** [5], **Xiaozhe Ma** [1,2,3,*] **and Leying Wu** [2,*]

[1] Key Laboratory of Geospatial Technology for the Middle and Lower Yellow River Region, Ministry of Education, The College of Geography and Environmental Science, Henan University, Kaifeng 475004, China
[2] Key Research Institute of Yellow River Civilization and Sustainable Development & Collaborative Innovation Center on Yellow River Civilization, Henan Province and Ministry of Education, Henan University, Kaifeng 475001, China
[3] Regional Planning and Development Center, Henan University, Kaifeng 475004, China
[4] State Key Laboratory of Earth Surface Processes and Resource Ecology (ESPRE), Beijing Normal University, Beijing 100875, China
[5] Institutes of Science and Development, Chinese Academy of Sciences, Beijing 100190, China
[*] Correspondence: mxz@henu.edu.cn (X.M.); 40150007@vip.henu.edu.cn (L.W.)

**Abstract:** Mitigating carbon emissions has become a pressing concern in the process of economic development across China due to China's key strategic goal of reaching peak carbon and carbon neutrality. Henan Province, which is located in the Central Plains, has less forest area and coverage than other areas of the nation, but consumes plenty of energy. Therefore, the quantification of Henan's potential carbon sink is crucial for the province's response to climate change due to the national commitment to carbon reduction targets. This research estimated the carbon sink of tree forest vegetation in Henan Province from 2019 to 2060 based on the CO2FIX model using data from the 9th Forest Inventory Report and the forest planning targets of Henan Province. The results show the following: (1) The carbon sink of existing arbor forests is mainly composed of ecological public welfare forests, and a small-year fluctuation in the carbon sink will result from the rotation of commercial forests. (2) The peak carbon sink years for existing forests and new afforestation are between the young and middle ages of the trees, and the peak of the carbon sink in Henan Province as a whole was in 2032. (3) More than 72.4% of the overall carbon sink in Henan Province's arbor forest vegetation comes from the above-ground portion. (4) The energy substitution effect of traditional and improved cookstoves is significantly enhanced during the main cutting period of the existing commercial arbor forest in Henan Province.

**Keywords:** vegetation carbon sink; CO2FIX model; new afforestation; forestry bioenergy; public welfare forest

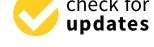



## 1. Introduction

The risks and impacts of climate change, a matter of global human survival, can be controlled through anthropogenic reductions in carbon emissions [1]. China is currently the largest emitter of anthropogenic carbon emissions and is facing enormous pressure and challenges to mitigate carbon emissions [2]. China announced in 2020 that it would aim to attain the peak of carbon emissions by 2030 and achieve carbon neutrality by 2060 to reduce emissions and develop an ecological civilization [3]. Working on both carbon sources and carbon sinks can help us achieve this goal [4]. Moreover, working to reduce anthropogenic carbon emissions needs to be complemented by increasing carbon sinks through natural solutions. Numerous studies have demonstrated that using nature-based solutions, such as forest carbon sinks, to solve these problems is more cost-effective and has numerous advantages [5–8].

Forests, the mainstay of terrestrial ecology, are essential to the carbon cycle and play a crucial role in preserving the carbon balance of terrestrial ecosystems [9,10]. Hence,

modeling and assessing forest carbon sinks have attracted much attention in studies related to mitigating carbon emissions. There are many international and domestic studies on carbon sinks, and at present, research on forest carbon sinks has become more advanced, relying on the rapid development of carbon sink modeling [11]. Basic research methods include sample plot inventories, model simulations, and remote-sensing estimation [12]. Most studies rely on models for carbon sink simulation, but some studies integrate multiple methods for forest carbon sequestration analysis. Zhang et al. [13] estimated the carbon sequestration of Shanghai's forests by combining typical sample plot inventory data with satellite images. Xu et al. [14] studied the carbon sink capacity of Shanghai's forests using typical sample plot inventory data combined with the CITYgreen model. Jia et al. [15] assessed the carbon sink potential of larch plantations in northern China using sample plot data and the CO2FIX model. Zhao et al. [16] studied the dynamics of carbon sequestration in Chinese forests from 1982 to 2019 based on remote sensing data and the FORCCHN model. The scope of research on forest carbon sequestration in China includes national, regional, provincial, and city levels. Fang et al. [17] simulated vegetation carbon sinks in China using terrestrial empirical data, remote-sensing images, and the conversion factor continuous function method. Zhang et al. [18] calculated the biomass and carbon sequestration in each region of China by establishing a regression model of subregional storage of biomass at the forest stand scale. Cheng et al. [19] have combined stockpile, biomass, and carbon stock to estimate the current status of carbon sinks in Gansu Province through forest inventory three-phase data. Some scholars also estimated forest carbon sink potential by subcountries and provinces. Ma et al. [20] used the CO2FIX model combined with data from the 6th forest inventory to estimate forest carbon sinks for 2003–2050 in each province of China, and Qiu et al. [21] used forest inventory data and sample plot data combined with empirical models to estimate forest carbon sinks from 2003 to 2050 for each province in China. Wang et al. [22] used empirical growth curves and some parameters to estimate the carbon storage and sequestration potential under the Grain for Green Program in Henan Province between 2000 and 2060. In addition, other scholars have focused on the study of forestry bioenergy due to its carbon mitigation effect. The world is rich in forest bioenergy resources [23]. Scholars, such as Chen et al. [24] and Yang et al. [25], have mentioned the important contribution of global forestry bioenergy to reducing carbon emissions when studying the carbon dynamics of forests and forest products. These scientific studies have provided important contributions to the modeling and analysis of forest carbon sequestration in China, but there are still uncertainties in assessing forest carbon stocks due to the research approach and the spatial and temporal heterogeneity of forests themselves [10,21].

The studies found that the carbon emissions from energy usage are higher in Henan Province. However, Henan Province's forests have a strong potential as a carbon sink because the forest area was 16.7 Mha in 2018, and the coverage rate was 24.14%, higher than the national average of 22.96% [6,26]. As the nation has prioritized achieving a carbon emissions peak and carbon neutrality, it is particularly important to estimate the carbon-sequestration potential of forests in Henan Province. At present, scholars have conducted relatively advanced studies on carbon emissions from energy consumption in Henan Province [27]; however, few studies have focused on carbon sinks, and those concerning forest carbon sinks are relatively limited. Moreover, these studies mainly use historical data, such as forest inventory and remote-sensing images [28–30], to study the dynamic changes in historical forest carbon stocks, and rarely predict the future trend of forest carbon sequestration in Henan Province. Even fewer studies have focused on the mitigation potential of forestry bioenergy in Henan Province.

Ecosystem process models play an important role in the simulation of forest carbon sinks. Compared to other simulations, the CO2FIX model can simulate the entire carbon cycle, from forest growth to forest product processing, and the simulation results are more comprehensive [15]. The CO2FIX model can simulate carbon dynamics in various age groups of different forest species and can be applied to various types of forest ecosys-

tems located in a wide range of climatic zones [31]. This model can also quantify the carbon-sequestration potential of afforestation projects [32], which can help in ecological construction and sustainable land-use planning [33]. Furthermore, in addition to calculating the reduction potential of forestry biomass energy sources, the model can mimic the carbon cycle in forest ecosystems. Therefore, this paper aims to investigate the dynamic trends of vegetation carbon sinks in Henan's arbor forests using the CO2FIX model, which can provide data support for Henan Province to cope with climate change. First, this paper calculates the vegetation carbon sink capacity of the existing arbor forest in Henan Province from 2019 to 2060 based on area and stock data from the 9th China Forest Resources Report, combined with the CO2FIX model. Second, based on the afforestation plan of Henan Province in the Forest Henan Ecological Construction Plan (2018–2027), this paper simulates the vegetation carbon sink potential of the newly planted arbor forest in Henan Province. Finally, this paper simulates the emission-reduction effect of forestry bioenergy substitution for fossil energy in Henan Province and analyzes the impact of biomass-combustion technology on the emission-reduction effect of bioenergy.

## 2. Methods and Data

### 2.1. Overview of the Study Area

Henan Province is located in central China, bounded by longitudes 110°21′–116°39′ East and latitudes 31°23′–36°22′ North. From south to north, Henan Province straddles the natural boundary between the warm temperate and subtropical zones [34], with two types of monsoon climate. The climate of Henan Province is characterized by hot and rainy periods, with high temperatures and relatively abundant precipitation in the summer and low precipitation in the winter. Temperatures are milder in parts of southern Henan and colder in the north–central region, with minimum temperatures below 0 °C.

Figure 1 depicts the topography of Henan Province. The topography of Henan Province is higher in the west and south than in the east–central and north, with a wide distribution of plains and basins, as shown in Figure 1. According to the ninth China Forest Resources Report, the forests of Henan Province are mainly distributed in the western Funiu Mountain region (the most extensive), the southern Tongbai Dabie Mountain region, and the northern Taihang Mountain ecoregion [35]. The area of arbor forest is 3.48 Mha, mainly consisting of Robur, poplar, and cypress species. The existing arbor forest is classified by age group structure, with the central portion being concentrated in young and middle-aged forests. The area and stocking volume of tree species gradually decline in the order of young, middle-aged, near-mature, mature, and over-mature forests. Contrary to the general trend, some dominant species, such as Chinese pine, are found to have a small percentage of the area of young and middle-aged forests. With 57.96% and 42.04% of the storage volume of public welfare forests and commercial forests, respectively, the acreage and storage volume of public welfare forests in Henan Province are marginally higher than those of commercial forests.

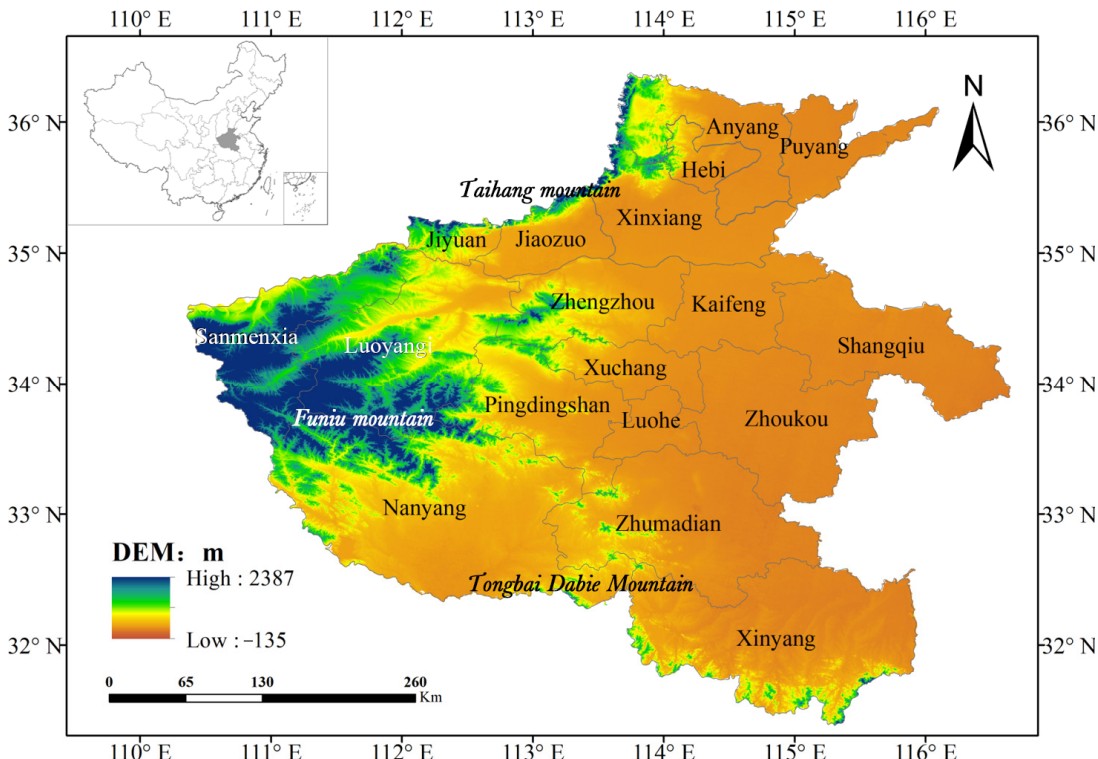

**Figure 1.** Topographical map of Henan Province.

*2.2. Methodology*

Carbon intensity is a crucial indicator for modeling carbon sinks in forest ecosystems. This paper uses the CO2FIX model to simulate the carbon density of different tree species through several investigations and assessments of carbon measurement models [36,37]. The Netherlands' Wageningen University created the CO2FIX model, which is based on a carbon-balancing model at the ecosystem level [15]. The carbon stock and carbon-flux dynamics of the forest biomass–soil–forest product chain are estimated at the stand level on an annual basis by setting up climate data, such as temperature and precipitation, in the CO2FIX model along with data such as felling cycles, organ growth change rates, and carbon content during tree growth [38]. The main sources of bioenergy from arbor forests in Henan Province are litter and abandoned wood products. It is possible to simulate the degree of carbon reduction, or 'carbon neutrality,' that would result from the replacement of fossil fuels with bioenergy from arbor forests and the improvement in biomass combustion efficiency in Henan Province by setting parameters such as the amount of biomass fuel (fuelwood) produced per year, the energy content of fossil fuels and bioenergy fuels, and the efficiency and emission factors of current and alternative technologies. Traditional and improved cookers are the two biomass-combustion technologies examined in this research. Research on the economics of carbon sinks has advanced due to the growing linkage between nature and economics [11]. The CO2FIX model has been enhanced regularly, and version V3.1 now includes financial and carbon accounting panels. The CO2FIX model V3.1's framework diagram can be found in Schelhaas et al. [39].

This paper's process of calculating carbon stocks and sinks is shown below.

$$C_t = \sum_i \sum_j CD_{i,j,t} \cdot S_{i,j,t} \tag{1}$$

where $t$ denotes the year; $i$ denotes the tree species; $j$ denotes the age group, including young, middle-aged, near-mature, mature, and over-mature forests; $C_t$ denotes the carbon stock of tree forest vegetation in Henan Province in the year $t$, and its unit is MgC; $CD_{i,j,t}$ denotes the carbon density of the age group $j$ of tree species $i$ in the year $t$, and its unit is

MgC/ha, as simulated by the CO2FIX model; $S_{i,j,t}$ denotes the area of the age group $j$ of tree species $i$ in the year $t$, and its unit is $10^2$ ha. The annual carbon sink of arbor forests in Henan Province is calculated as shown in Equation (2).

$$CS_t = C_t - C_{t-1} \tag{2}$$

where $CS_t$ denotes the carbon sink in the year $t$ and its unit is Mg; $C_t$ denotes the carbon stock in the year $t$, and its unit is MgC; $C_{t-1}$ denotes the carbon stock in the year $t-1$, and its unit is MgC.

*2.3. Data*

Based on the existing forest inventory data and afforestation targets in Henan Province, this paper investigated the carbon sequestration status of arbor forest vegetation in the province. The data on the area and storage volume of each age group of dominant tree species in Henan Province can be obtained from the China Forest Resources Report 2014–2018. The existing arbor forests in Henan Province are divided into public welfare forests and commercial forests based on five forest species, and the carbon sink capacity of those forests is calculated in this study. Public welfare forests mainly consist of protection and special-purpose forests, whose primary function is ecological protection and restoration. In contrast, commercial forests are timber, charcoal, and economic forests, where trees are rotated during the growth process to generate economic value. In addition to calculating the carbon sink of existing arbor forests in Henan Province, this paper also predicts the carbon sequestration status of new afforestation in Henan Province.

The selection of tree species is one of the crucial issues in carrying out carbon sink modeling of arbor vegetation. As indicated in Table 1, the six dominant tree species in Henan Province, with dominant area and stocking volume among the existing arbor forest species, were chosen for this study based on the ninth forest inventory report and the research of academics such as Ma et al. [40] and Xu et al. [41]. The larger area of tree species indicates that they easily survive in Henan Province and are more adaptable to the climatic conditions of Henan Province, while the forest stock refers to the total volume of tree trunk wood growth in the forest area, which can reflect the total scale and level of forest resources in a country or region. Therefore, the dominant tree species selected in this paper in Henan Province arbor forest are poplar, broad-leaved mixed forest, Robur, Masson pine, Chinese pine, and cypress. These six tree species make up the top six species in terms of arbor forest area, and their combined storage volume makes up 85.53% of the arbor forest storage volume in Henan Province. The carbon-sink simulation of these six dominant tree species can fully illustrate the current status and evolution of the carbon sequestration trend of arbor forest vegetation in Henan Province.

**Table 1.** Accumulation and area of dominant tree species in Henan Province.

| Serial Number | Dominant Tree Species | Accumulation (10⁴ Cubic Meters) | Accumulation Ratio (%) | Area (10⁴ ha) | Area Ratio (%) | Public Welfare Forests Area (10⁴ ha) | Commercial Forest Area (10⁴ ha) |
|---|---|---|---|---|---|---|---|
| 1 | Poplar | 6280.42 | 30.31 | 62.36 | 17.91 | 38.38 | 23.98 |
| 2 | Broad-leaved mixed forest | 5701.66 | 27.52 | 106.37 | 30.54 | 65.46 | 40.91 |
| 3 | Robur | 4450.57 | 21.48 | 77.83 | 22.35 | 47.90 | 29.93 |
| 4 | Masson pine | 701.58 | 3.39 | 10.50 | 3.01 | 6.46 | 4.04 |
| 5 | Chinese pine | 352.69 | 1.70 | 6.60 | 1.90 | 4.06 | 2.54 |
| 6 | Cypress | 234.94 | 1.13 | 10.68 | 3.07 | 6.57 | 4.11 |

Modeling the interannual carbon dynamics of arbor forests is another critical issue in estimating carbon sink potential, which was obtained through the CO2FIX model. In this study, *.CO2 data files on individual tree species were constructed based on the CO2FIX model, after which CO2FIX's parameters were configured for carbon flux simulation, and lastly, the files were exported in text or Excel format. The specific parameters for each module were set as follows, and the climate data in the soil module of the CO2FIX model

were taken from the World Climate website (http://www.worldclimate.com/, access date: 15 November 2022). In this paper, the mean values of the multicity climate data of Henan Province were set as the parameters of the soil module. The initial soil carbon stock in the soil module was set to zero because this study focused on the vegetation carbon sink of the arbor forest and did not focus on the soil carbon pool. The parameters, such as efficiency and emission factors for current and alternative technologies, in the bioenergy module, were taken from Ma et al. [7], which assessed the emission reduction effect of forestry bioenergy based on the actual situation in China. The tree growth data in the biomass module and the parameters in the products module were taken from Ma et al. [40]. This paper obtained parameters such as CAI and growing period of dominant tree species in China through sample plot data and extensive literature research. It is important to note that this paper calibrates the carbon content of each organ of the dominant trees in the CO2FIX model. In previous studies, most authors used a fixed average carbon content value of 0.5, failing to consider the effect of tree type and the different organs in the tree on the carbon content [42]. In this paper, the carbon content of different organs was improved in the CO2FIX model based on the findings of Wang et al. [43] and Ma et al. [7] in China. The key parameters are summarized in Table 2.

**Table 2.** Carbon content of different organs of dominant tree species in Henan Province.

| Dominant Tree Species | Stems (%) | Roots (%) | Foliage (%) | Branches (%) |
|---|---|---|---|---|
| Poplar | 46.09 | 43.76 | 44.8 | 46.13 |
| Broad-leaved mixed forest | 45.36 | 43.68 | 45.61 | 45.39 |
| Robur | 44.34 | 42.26 | 45.86 | 44.73 |
| Masson pine | 47.91 | 45.96 | 49.8 | 48.32 |
| Chinese pine | 47.36 | 45.92 | 49.68 | 48.3 |
| Cypress | 48.33 | 46.13 | 50.53 | 48.31 |

Note data sources [7].

The annual new afforestation area is one of the crucial indications when calculating the carbon-sink potential of new arbor forests. This paper equitably divides the area of new forests into years and tree species by the afforestation plan target of Henan Province in the Forested Henan Ecological Construction Plan (2018–2027), which states that the forest cover in Henan Province will reach 27.56% by 2022 and 30% by 2027. The total new planted area of 829,700 ha in the afforestation plan of Henan Province for the period of 2018–2022 is equally divided into 2019–2022, with an additional area of 34,600 ha per year for each dominant tree species; the total afforestation area was 671,700 ha from 2023 to 2027, with an additional area of 222,400 ha per year for each dominant tree species; the total afforestation area from 2028 to 2030 is based on the expectation of forest cover in Henan Province, with a minimum standard of 31% forest cover by 2030. The new arbor forest area is estimated to be 167,000 ha by 2030, and the new area of each dominant tree species is estimated to be 9300 ha per year. The area of arbor forest in Henan Province is set in this study to stay constant from 2031 onwards, in line with the area in 2030, because the forest-planning targets for 2031–2060 are not yet available.

### 3. Results and Analysis

*3.1. Vegetation Carbon Sinks of an Existing Arbor Forest*

This paper illustrates the vegetation carbon sequestration of existing arbor forests in Henan Province in two parts. The existing arbor forests refer to the arbor forests already planted in Henan Province at the time of the 9th forest inventory. To clearly depict the vegetative carbon content of an existing arbor forest, this section has been divided into two parts. The first part presents the vegetation carbon sink of public welfare forests and commercial forests in existing tree forests in Henan Province. The second part describes the carbon stocks of six dominant tree species in Henan Province. In particular, it should be noted that the carbon stock of the six dominant species simulated in this paper does

not represent the total vegetation carbon stock of the entire arbor forest in Henan Province. Therefore, this paper divided the total vegetation carbon stock of the six dominant tree species by their share in the total arbor forest volume (85.53%) to obtain the total vegetation carbon stock of the existing arbor forest in Henan Province, as shown in Figure 2.

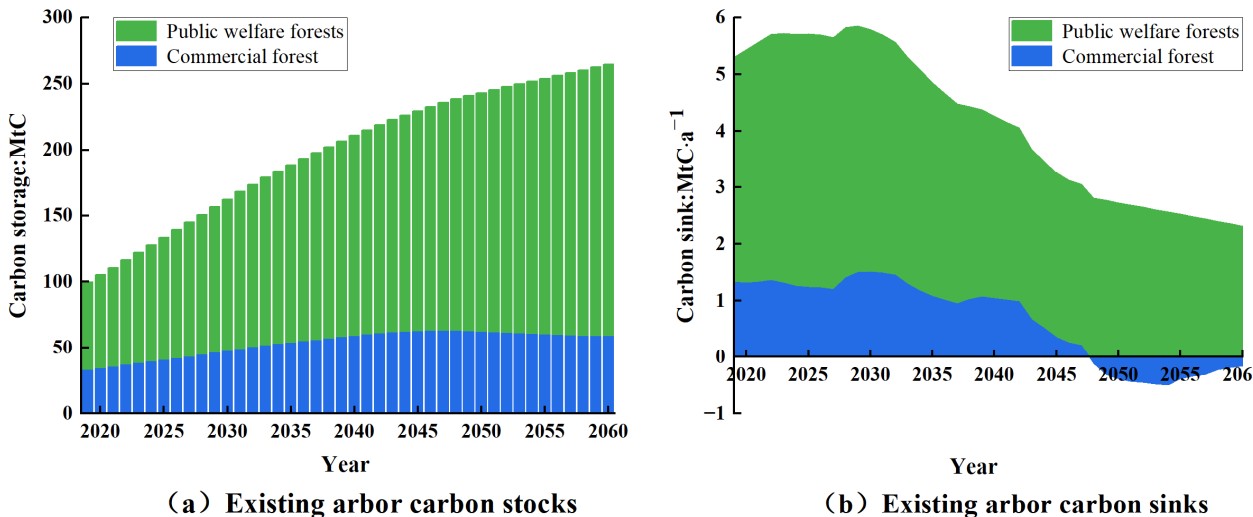

（a）**Existing arbor carbon stocks**　　　　　　　（b）**Existing arbor carbon sinks**

**Figure 2.** Carbon-sink capacity of existing arbor forest vegetation in Henan Province, 2019–2060.

　　　The long-term accumulation and the short-term significant loss of forest biomass can lead to the conversion of forests into carbon sources and sinks [44]. The carbon stock of existing arbor forests in Henan Province is shown in Figure 2a, and the annual carbon sink is shown in Figure 2b. As seen in Figure 2, the carbon stocks increase year by year and slowdown in the later part of the simulation. The average annual carbon sink reaches 4.16 MtC/a during the modeling period. The annual carbon sink rises annually at the beginning of the simulation so that the annual carbon sink of existing forests reaches a peak of 5.85 MtC/a in 2029 but declines rapidly thereafter. This is because existing arbor forests are currently mostly in their young to middle-aged stage, when their capacity to sequester carbon is strong, whereas, by 2029, existing arbor forests will start to enter their mature and over-mature stage, which is when their capacity to sequester carbon begins to slightly decline. At the same time, the vegetation carbon density increases from 28.64 MgC/ha at the start of the simulation to 76.04 MgC/ha in 2060, which is related to the consistent increase in carbon stocks in existing arbor forests. The trend of vegetation carbon stock in public welfare forests is consistent with that of the total arbor forest from 2018 to 2060, with an average annual growth rate of 2.79%. The carbon stocks of commercial forests in Henan Province first increase and then decrease gently, and carbon sinks have interannual fluctuations that are closely related to the rotation cycle of commercial forests. During the main cutting period of commercial forests, the carbon density of each dominant tree species decreases dramatically, thus, leading to the change in commercial forests from carbon sinks to carbon sources.

　　　An overview of carbon sequestration in arbor forests is given above, and the description of vegetative sequestration by tree species is given below. Figure 3 displays the ability of the leading tree species in the public welfare forest of Henan Province to store carbon. As the number of years increases, the carbon-sink capacity of all existing dominant tree species in Henan Province, except Robur, tends to decrease slightly during the simulation period. The carbon sink capacity of the dominant tree species in Henan Province is closely correlated with their carbon density and area before and after tree maturity. The dominant species' carbon density increases quickly during the young and middle stages of the trees' development, slowing down after maturity, which is consistent with the findings of previous studies [45,46]. The dominant tree species, broad-leaved mixed forest, Robur, and cypress, have a relatively large area of medium and low forest age, greater than 89.51% of

the total area, while the remaining species have a lower percentage. This explains why these three species' carbon stocks rise more quickly and rapidly in young and medium-sized forests and why the capacity of carbon sinks exhibits a more pronounced upward trend that is not found in the remaining species.

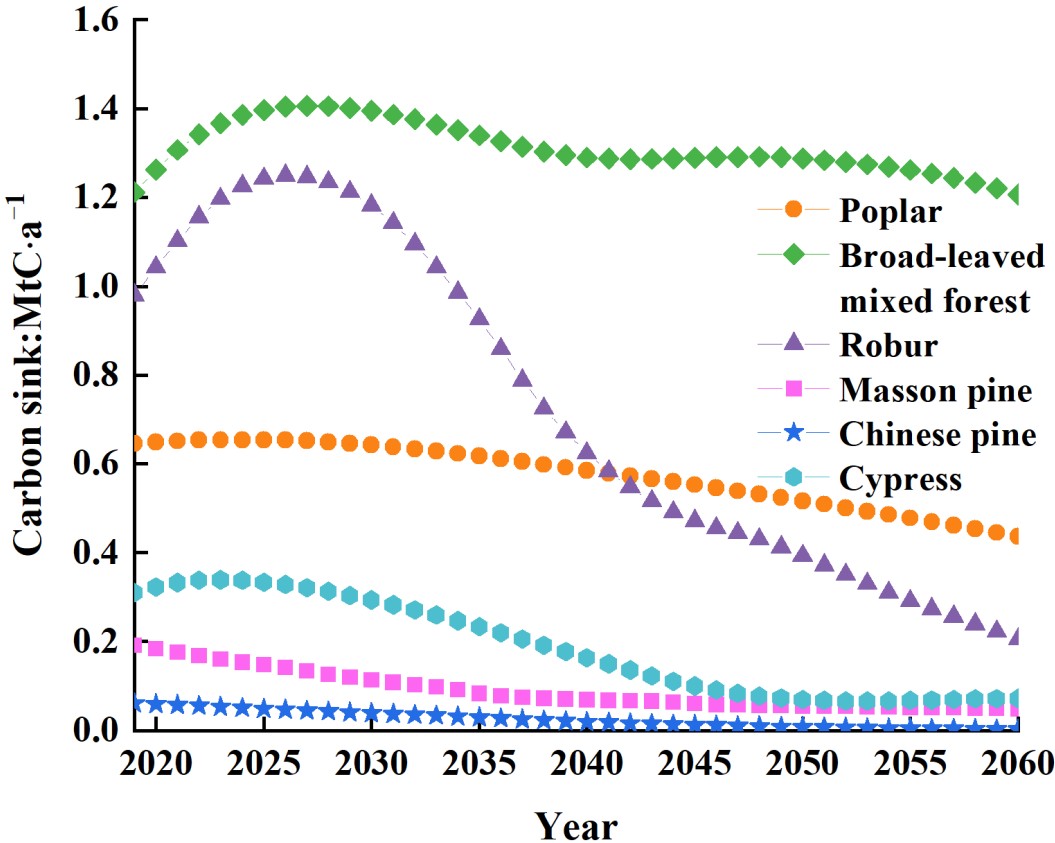

**Figure 3.** Carbon sinks in public welfare forests of dominant tree species in Henan Province, 2019–2060.

Figure 4 illustrates the annual carbon-sequestration potential of the dominant tree species in commercial forests in Henan Province. The annual carbon-sequestration capacity of each dominant tree species fluctuates during the simulation period, and the carbon sink capacity decreases over the main logging period, divided into three cases. The first is Masson pine, which has a good carbon sink capacity throughout the simulation period. This is mainly due to the high carbon density and low variability, which are affected less by logging. The second scenario is a change from a carbon sink to a carbon source during the main logging period, where broad-leaved mixed forest, Robur, and cypress have a large proportion of young and medium-aged areas, a rapid increase in carbon density, and carbon sink capacity before the main logging period. The third type is the change from a sink to a carbon source both before and after the main harvesting period. Poplar and Chinese pine are more significantly affected by logging due to the lower carbon density during the simulation period and the small area of the tree species until near maturity.

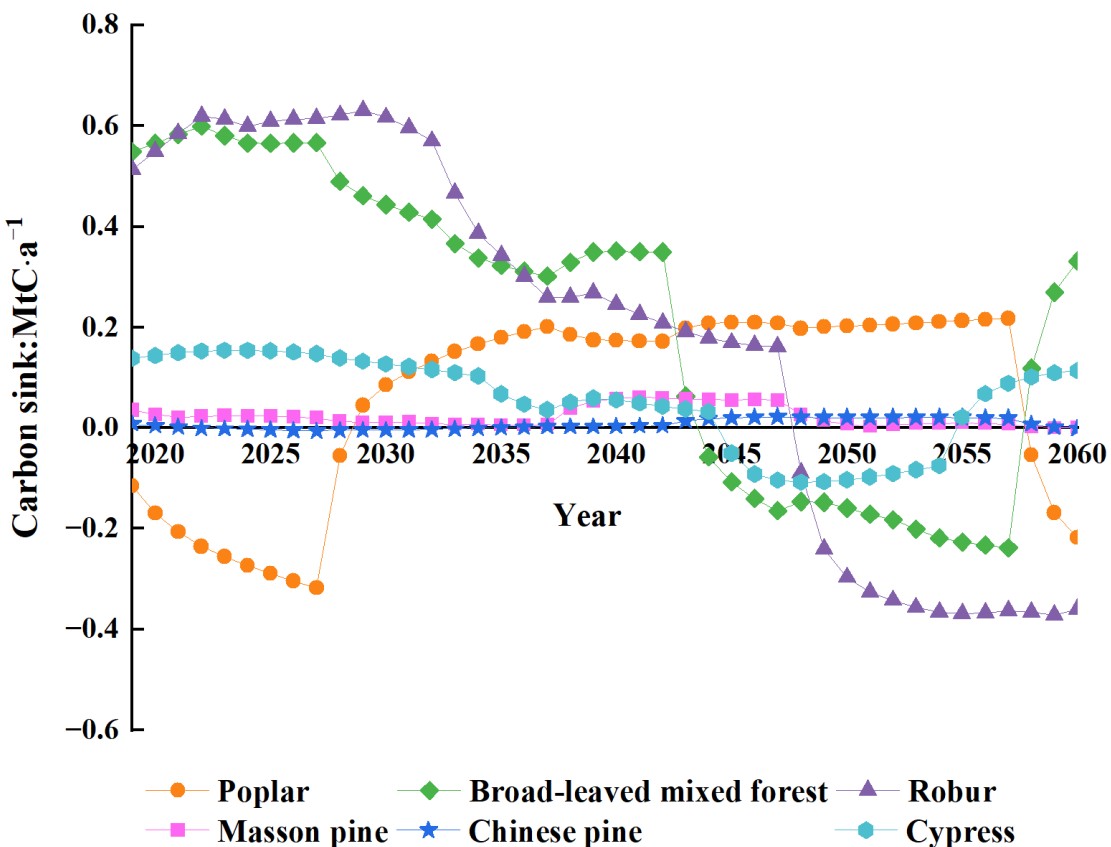

**Figure 4.** Carbon sinks in commercial forests of dominant tree species in Henan Province, 2019–2060.

*3.2. Vegetation Carbon Sinks of New Afforestation*

According to the Henan Forest Ecological Construction Plan (2018–2027), the focus of forestry development in Henan Province is on ecological construction. Thus, this article regards all new afforestation as public welfare forests, which are conducted to simulate the carbon sink potential of public welfare forests in meeting the planning objectives.

Figure 5 displays the carbon density and annual vegetation carbon sink of newly planted arbor forests in Henan Province. It can be seen in Figure 5 that the vegetation carbon density of the new afforestation has been increasing in line with the trend of carbon stock throughout the simulation period. The average annual change in the carbon density is 13.13%, ranging from 0.54 MgC/ha at the start of the simulation to 80.6 MgC/ha in 2060. The rate of increase in vegetation carbon density was slow in both the early and late stages of the simulation, but for different reasons. In the early stages, the area of newly planted trees was limited; in the later stages of the simulation, the rate of carbon sequestration slowed down when each newly planted tree was in the mature or over-mature stage. In the middle of the simulation, the afforestation plan was completed, and the forest area in Henan Province increased significantly. Meanwhile, the newly planted forests gradually grew into middle-aged forests, so the annual carbon sink of the new afforestation reached a peak of 5.36 MtC in 2038. This is the reason why the annual carbon sink of newly planted forests in Henan Province tends to increase and then decrease. These trends are consistent with the findings of previous studies on carbon stock in natural forest reserves [46].

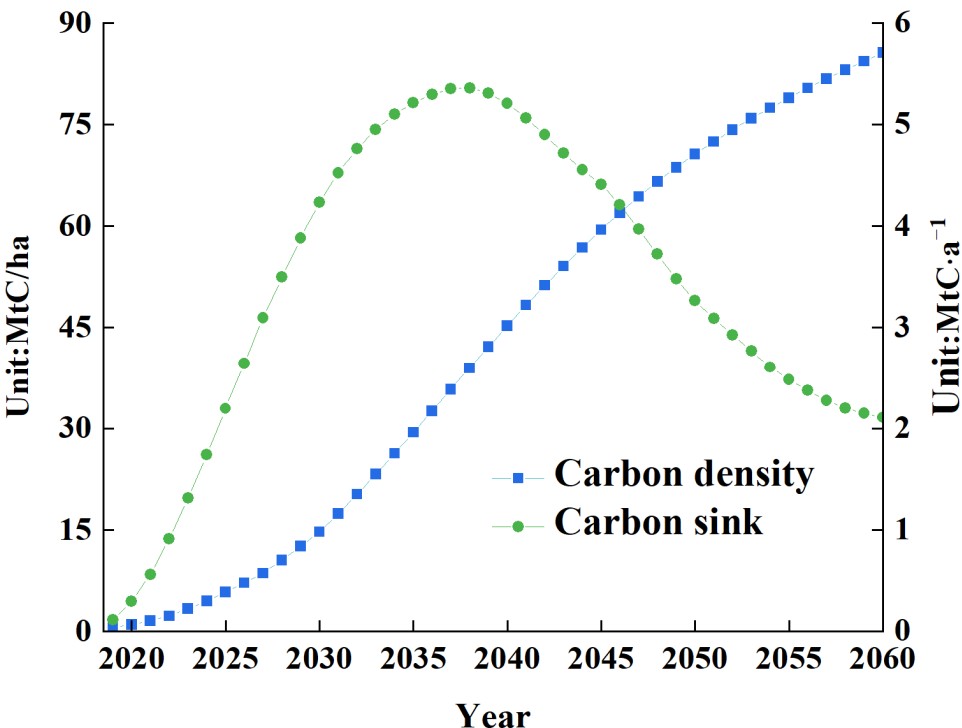

**Figure 5.** Carbon density and carbon sink of new afforestation in Henan Province, 2019–2060.

Figure 6 displays the annual vegetation carbon sinks of newly planted dominant tree species in Henan Province. The annual carbon sinks of these dominant tree species show an inverted U-shaped trend, and the peak carbon sink years of the dominant tree species in new plantations were those of trees that were young or of middle age. Cypress is the largest contributor to the carbon sink of new plantations, and Chinese pine is the lowest. According to the forest-planning objectives of Henan Province, the new forest area in each stage of forest planning was equally distributed amongst each dominant species. Therefore, the main influencing factor of carbon sink in newly planted forests is the carbon density of dominant species. Cypress has the highest carbon density of the dominant species, while poplar and Chinese pine have the lowest carbon densities, respectively. It should be noted that the carbon sink of Chinese pine and poplar changes at different growth stages of the trees. In the early stages of tree growth, the carbon density of poplar is greater than that of Chinese pine, while in the later stages, the opposite is true; this conclusion is also consistent with other scholars' studies [47,48]. As shown in Figure 6, cypress and Masson pine have higher carbon sinks in the same area, while the rest of the dominant species have lower ones. The existing dominant species in Henan Province are broad-leaved mixed forest, Robur, and poplar in a large area, while cypress and Masson pine have a limited forest area. Based on the perspective of the carbon intensity of the dominant species, the planting of cypress and Masson pine should be increased in the future in order to better achieve the-carbon reduction target of Henan Province.

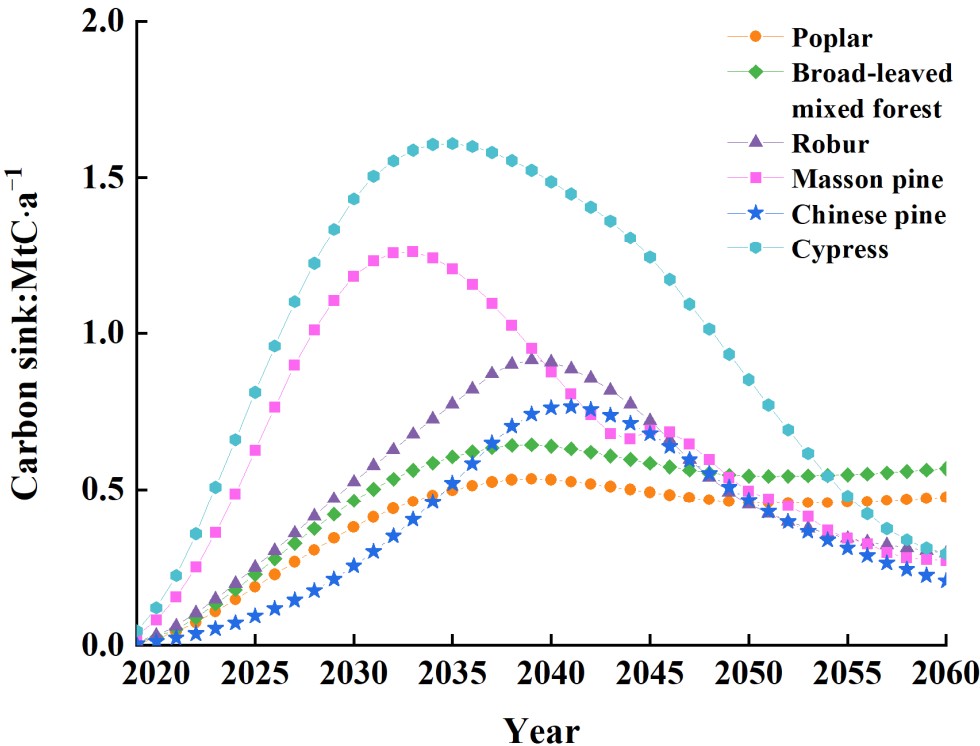

**Figure 6.** Carbon sinks of dominant tree species for new afforestation, 2019–2060.

### 3.3. Vegetation Carbon Sinks of All Arbor Forests

The previous section illustrated the vegetation carbon sink of existing and newly planted arbor forests in Henan Province. Combining the two gives the total carbon sink of arbor forest vegetation in Henan Province from 2019 to 2060. This section describes the vegetation carbon sink of all arbor forests in Henan Province from two perspectives. Initially, the carbon stock of all arbor forests' vegetation at various modeling stages is analyzed; finally, the carbon stock of all arbor plantations is described in two parts, above and below ground. The above-ground component includes carbon sequestered by tree trunks, leaves, branches, and forest products, while the below-ground component includes the roots of trees and carbon that sank into the soil.

Figure 7 illustrates the overall arbor forest carbon density and annual carbon sink in Henan Province from 2019 to 2060. The carbon density of arbor forest vegetation in Henan Province shows an increasing trend during the simulation period, but the growth rate slows down in the later part of the simulation, which is consistent with the trend of carbon stock throughout the simulation period. The annual vegetation carbon sink of the arbor forest in Henan Province increased significantly at the beginning of the simulation due to two dominant factors. On the one hand, this was due to the annual increase in the area of new forest plantings, and on the other hand, this was because the existing forests were at the middle and young age stage, which is when forests have the highest carbon-sinking capacity. In the later part of the simulation, the annual vegetation carbon sink decreases, partly because the forest area does not increase after 2030 and partly because commercial forests enter the harvesting cycle. Before 2030, existing arborvitae in Henan Province accounted for the majority of the arbor forest's carbon sink. However, after 2030, newly planted arbor forests accounted for the plurality, contributing 53.59% of the overall arbor forest carbon sink. The average annual carbon sink of an arbor forest in Henan Province is 7.64 MtC/a, with a peak in 2032, which can sequester 10.32 MtC/a.

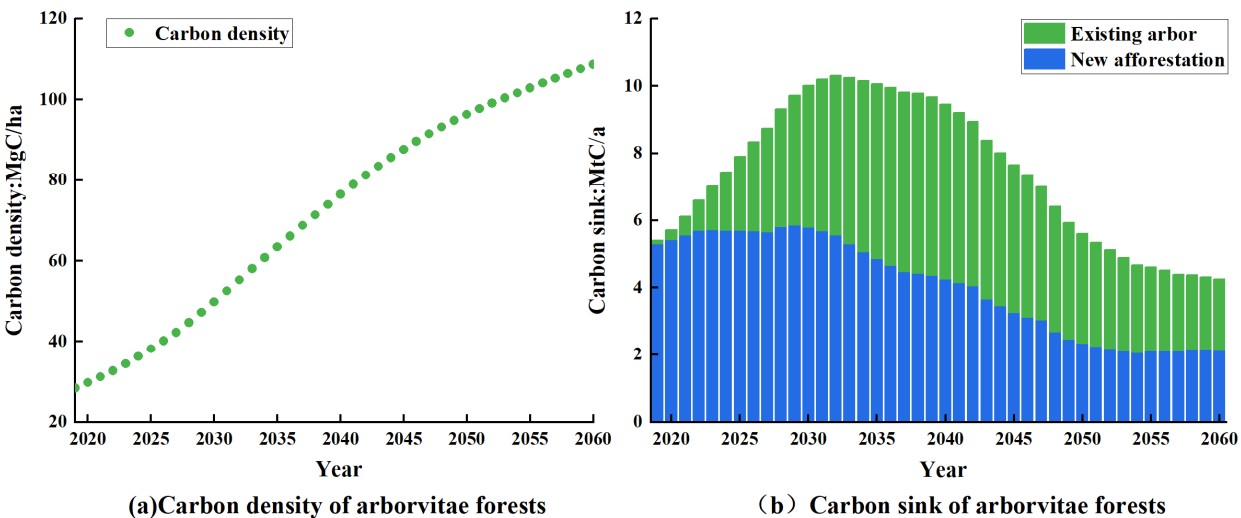

**Figure 7.** All arborvitae forests' carbon density and carbon sink in Henan Province, 2019–2060.

The carbon storage and carbon sink of the above-ground and below-ground parts of the tree forest vegetation in Henan Province are shown in Figure 8. During the growth of arbor forests in Henan Province from 2019 to 2060, carbon storage was mainly concentrated in the above-ground part, and its carbon density remarkably rose during the simulation period. The above-ground portion of the carbon stock has a greater average annual growth rate of 3.58% than the below-ground portion, 3.2%. The above-ground fraction is also the main component of the carbon sink of forest vegetation in Henan Province, contributing at least 72.4% overall. This is because the tree's biomass is mainly concentrated in the stems, branches, and foliage, and the carbon content of the stems, branches, and foliage is higher than that of the roots. In addition, the cumulative carbon sequestration in the soil during the simulation period was 61.4 MtC, and 18.2% of the carbon sink of forest vegetation was stored in the soil. The relatively small amount of vegetation carbon sinks contained in the soil is due to the relatively limited amount of apomictic material from the growth of trees. The ratio of carbon sinks in roots to carbon sinks in the soil in the below-ground part is approximately 1:2, as shown in Figure 8b. The relatively small amount of vegetation carbon sink contained in the soil is due to the relatively limited amount of apoplankton produced during the growth of trees.

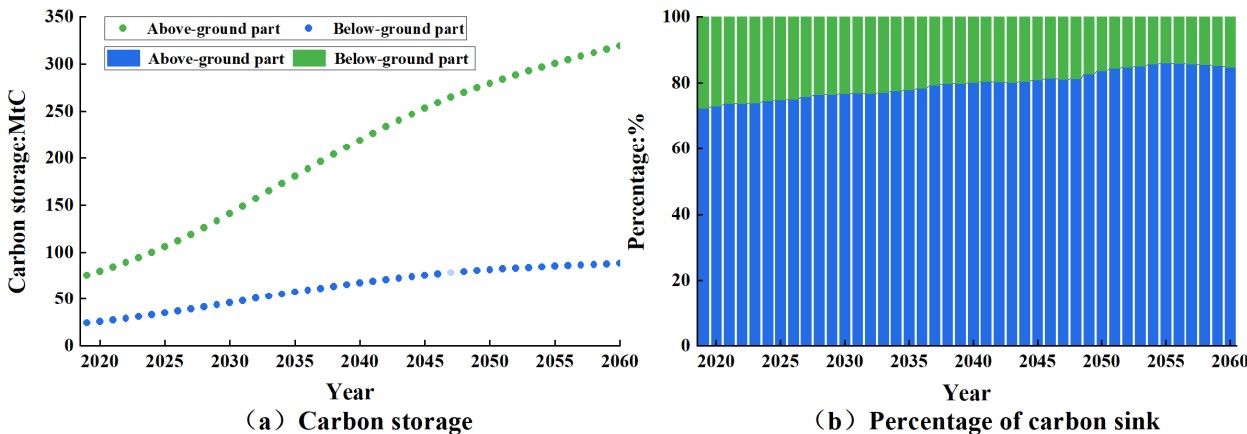

**Figure 8.** Carbon storage and carbon sinks in above- and below-ground parts of arbor forest vegetation in Henan Province, 2019–2060.

*3.4. Emission-Mitigation Effects of Arbor Forest Bioenergy*

The harvesting and recycling of waste forest products are the main sources of forest bioenergy in Henan Province. This article assumes that there is no logging in public welfare forests, and thus forestry bioenergy comes mainly from commercial forests. As bioenergy is carbon neutral, it is a perfect substitute for fossil energy and has a specific carbon reduction effect [25]. Therefore, this paper explores the carbon-abatement effects of forestry bioenergy in Henan Province under two bioenergy-combustion technologies: a traditional cookstove and an improved cookstove. This section first assesses the carbon mitigation effect of forestry bioenergy under these two technologies, which refers to the use of forestry bioenergy to replace coal; then, the carbon sequestration of arboreal vegetation is combined with the bioenergy-mitigation effect to analyze the integrated mitigation potential of arboreal vegetation.

Figure 9 demonstrates the cumulative carbon emissions from the arbor forestry bioenergy at two different combustion efficiencies The carbon reduction from the improved forestry bioenergy cookstoves is more than twice that of traditional cookstoves, as reflected by average annual emission reductions of 0.53 MtC/a and 1.07 MtC/a for conventional and improved cookstoves, respectively. From 2018 to 2060, the emission-abatement effect of arboreal forest biomass in Henan Province increases, which can be divided into two phases according to the rate of increase. From 2019 to 2042, trees are in their growth phase, and very few trees are logged, which results in less forestry bioenergy being obtainable. In this situation, the cumulative emission reductions obtained from using traditional cookstoves and improved cookstoves amount to 9.77 MtC and 20.09 MtC, respectively. Between 2043 and 2060, when most of the trees enter the main harvesting period, a large amount of felling residues will be generated to be burned as fuel, and the emission reduction of forestry bioenergy will be significantly enhanced. In this situation, 11.99 MtC of carbon could be reduced by 2060 using traditional cookstove technology, and 23.88 MtC of carbon could be reduced by 2060 using improved cookstove technology with higher combustion efficiency, which increases the average annual carbon reduction for 2019–2042 by 0.24 MtC/a and 0.45 MtC/a, respectively.

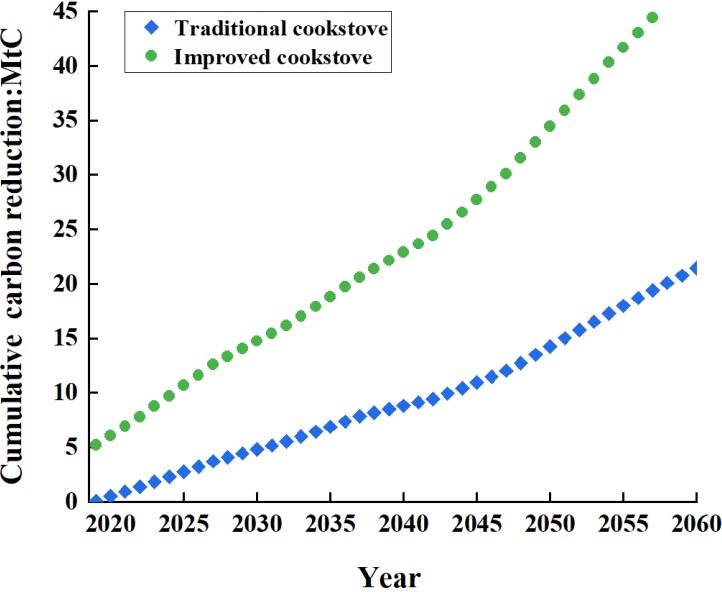

**Figure 9.** Cumulative bioenergy reduction from traditional and modified cookstoves in arboreal forests in Henan Province, 2019–2060.

Comprehensively taking into account the sink-enhancement effect of forest vegetation and the emission reduction effect of forestry bioenergy, the entire emission-reduction scale of arboreal forest vegetation in Henan Province in 2019–2060 is shown in Figure 10.

Compared with no consideration of bioenergy emission reduction, the scale of carbon reduction from vegetation sequestration in arboreal forests is slightly larger when both traditional and improved cooking technologies are used. The following are descriptions of the three scenarios: First, the replacement effect of bioenergy is not considered; the cumulative carbon sink of arbor forest vegetation in Henan Province is 313.17 MtC from 2019 to 2060, peaking at 10.32 MtC/a in 2032. Second, considering the substitution effect of traditional cookstove technology, the total carbon reduction is 334.92 MtC, peaking at 10.69 MtC/a in 2033, while bioenergy accounts for 6.5% of the cumulative carbon sink. Finally, after considering the substitution effect of improved cookstove technology, the total carbon reduction is 357.14 MtC, peaking at 11.11 MtC/a in 2033, while bioenergy accounts for 12.31% of the cumulative carbon sink. Based on the above three contexts, it seems that to achieve the 'double carbon target' as early as possible, the forestry sector in Henan Province should pay attention to the use of forestry bioenergy, especially commercial forest bioenergy. In addition, the efficiency of bioenergy utilization should be improved to further enhance the carbon sequestration potential of the forestry sector.

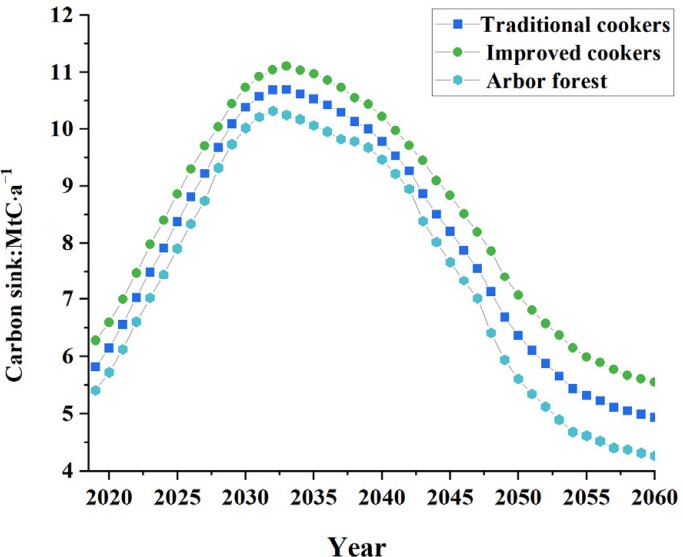

**Figure 10.** Carbon sink of tree forest vegetation in Henan Province under the replacement effect of bioenergy, 2019–2060.

## 4. Discussion and Conclusions

### 4.1. Discussion

Due to the spatial and temporal heterogeneity of forests, there is uncertainty in predicting future carbon sinks based on existing data. Qiu et al. [21] modeled the forest carbon stock of China's provinces from 2003 to 2050. Wang et al. [22] estimated the carbon storage and sequestration potential in Henan Province under the Grain for Green Program between 2000 and 2060. By comparing the results of the above articles with this paper, it was found that Henan Province has a great potential for carbon sequestration during the period 2020–2050. Qiu et al. [21] estimated the net carbon sequestration in Henan Province to be 53.39 MtC, and Wang et al. [22] estimated that the net carbon sequestration in Henan Province under the Grain for Green Program is 82.92 MtC. This article estimates carbon sequestration for 2020–2050 at 261.23 MtC. To explain this difference, the following analysis was conducted.

Qiu et al. [21] conducted a five-year continuous phase projection based on the 8th forest inventory data; the area of new afforestation in each province and region is divided according to the proportion of the area of tree species in China's 8th forest inventory. However, the area of new plantations in this paper is evenly distributed among the dominant tree species, which results in a high carbon sink for new plantations. For example, the

carbon density of cypress was relatively higher in Henan Province. Due to the average distribution of new plantation areas, the area of cypress in this paper is much larger than that of Qiu et al. [21]. Therefore, the overall carbon density of tree forests in this paper is higher than that of Qiu et al. [21].

Wang et al. [22] used the planted area of various tree species under the Grain for Green Program of Henan Province from 2000 to 2012 and the growth equation of the trees to simulate the potential for carbon sequestration from 2000 to 2060. The effect of newly planted trees on the carbon sequestration capacity was not considered. In contrast, our study considers both existing and newly planted tree forests in Henan Province, and the area of tree forests is much larger than that of Wang et al. [22], so the simulation results in this paper are higher than Wang et al.'s [22] findings.

In this study, the CO2FIX model is used to simulate the carbon sink of arbor forest vegetation in Henan Province for a long time period, but the parameters such as climate remain the same throughout the simulation period, which introduces uncertainties in the simulation results [49]. The total area of newly planted forest in this study is evident but averaged across the dominant species, which means that the area of new tree plantations set in this study may deviate from the actual area, thus, creating some uncertainty.

There are numerous ways to calculate the carbon sink potential of forests, and each method has advantages and disadvantages. However, it is worth acknowledging that the CO2FIX model has great advantages in carbon flux simulation and can be used to simulate the carbon cycle in forest ecosystems and estimate the reduction potential of forestry bioenergy. At present, the forest area of Henan Province is mainly composed of middle and young age groups, which have a huge carbon sink potential. We hope that the results of the carbon sink potential simulation in this paper will help Henan Province achieve the 'double carbon target'.

### 4.2. Conclusions

The above research has led to the following five conclusions:

(1) The carbon sink potential of arbor forests in Henan Province between 2019 and 2060 is 313.17 MtC, including existing and new tree forests, whose carbon sinks change in an inverted U-shaped curve during the simulation period, peaking at 10.32 MtC/a in 2032. The main contributors of carbon sinks are public welfare forests, and commercial forests have harvesting periods, which explain the small yearly fluctuations in carbon sinks in Henan's tree forests. In addition, some of the dominant species in commercial forests may change from carbon sinks to carbon sources during the main harvesting period.

(2) The carbon density and area of the dominant tree species in the arbor forest determine their carbon sequestration capacity. After the CO2FIX model simulation, it was found that the carbon density of dominant tree species increases rapidly when they are of middle and young ages, and the increase in carbon density slows down after maturity. Therefore, for existing arbor forests, when the existing planting area and carbon density of tree species are dominant, their carbon sequestration capacity is strong, and the dominant tree species, broad-leaved mixed forest, is the main contributor to its carbon sink. For new afforestation, cypress has the largest carbon density due to the new area added in this article, and it is a major contributor to the carbon sink of new arbor forests.

(3) The carbon sink produced by an arbor forest's above-ground portion (tree trunks, leaves, branches, and forest products) is more than 72.4% of the total carbon sink. Although the trend of below-ground (tree roots and soil) carbon sink growth is more moderate, it cannot be disregarded, particularly in the soil component. Nearly one-fifth of the carbon sinks of all arbor forest vegetation between 2019 and 2060 are stored in the soil.

(4) When considering the carbon-reduction effect of forestry bioenergy in Henan Province, this paper finds that improved cookstoves produce twice as much carbon reduction as traditional cookstoves. Combining vegetation carbon sequestration and biomass emissions reduction, the peak carbon sequestration and emissions reduction year for both cooker-burning technologies is 2033. In addition, the carbon emission abatement effect of

traditional and improved cookstoves is significantly increased after the commercial forest enters the main harvesting period. The peak carbon reduction year for conventional cooker-burning technology is 2051, with a reduction of 0.76 MtC, while the peak carbon reduction year for improved cooker-burning technology is 2054, with a reduction of 1.47 MtC.

(5) To better achieve the 'double carbon target' of Henan Province, this paper provides a reference for the forestry sector of Henan Province from the perspectives of new afforestation and bioenergy utilization. The first is new afforestation; based on the carbon intensity of the dominant tree species, Henan Province can consider increasing the planting area of cypress and Masson pine. Then, from the perspective of biomass energy, special attention should be paid to bioenergy in commercial forests. In addition, the efficiency of bioenergy utilization should be improved.

**Author Contributions:** Conceptualization, K.C.; methodology, K.C. and X.M.; validation, K.C. and X.M.; formal analysis, K.C.; writing—original draft, K.C. and X.M.; writing—review and editing, K.C., J.W., X.M. and L.W. All authors have read and agreed to the published version of the manuscript.

**Funding:** This research was funded by the National Natural Science Foundation of China (No. 41701632, 41871219, and 41901239), the major project of the Collaborative Innovation Center on Yellow River Civilization jointly built by Henan Province and the Ministry of Education (2020 M19), and Scientific Promotion Funding of the Prioritized Academic Discipline (Geography, Henan University).

**Data Availability Statement:** Forest group and age group data for tree species are from the published "China Forest Resources Report 2014–2018"; data on the area of new afforestation is available on a publicly available website: https://www.henan.gov.cn/2018/09-21/692208.html access date: 15 November 2022; The parameters, such as efficiency and emission factors for current and alternative technologies, in the bioenergy module, were taken from Ma et al. [7], the tree growth data in the biomass module and the parameters in the products module were taken from Ma et al. [40]; the climate data in the soil module of the CO2FIX model are taken from the World Climate website: http://www.worldclimate.com/ access date: 15 November 2022; the carbon content of different organs are based on the findings of Wang et al. [43] in China.

**Conflicts of Interest:** The authors declare no conflict of interest.

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
