# Peer review of "Simulation of Carbon Sink of Arbor Forest Vegetation in Henan Province of China Based on CO2FIX Model"

_land, doi:10.3390/land12010246_

Round 1

Reviewer 1 Report

Please refer to the attached document.

Reviewer 2 Report

The paper simulated the forestry carbon sinks in the Henan Province of China by using the popular CO2FIX model. The study is of significance in using a computer model to quantify the current and future forestry carbon sink capacities of the province. The simulation results can help guide long-term policymaking. The study fills the literature gap in carrying out experimental data. 

The paper is structured well; the methodology is clear; the results were discussed thoroughly; the references are rich and well organized. It is recommended to publish after the following issues are addressed:

1) please add unit for the DEM in Figure 1, also draw out the highlighted forestry regions, such as Funiu Moutain, Tongbai Dabie Mountain, and the northern Taihang Moutain eco-region, so the readers can follow the narrative well

2) Please name the studies referenced in the paper instead of simply numbered, such as line 163, line 208, line 216

3) For figure2, needs to clearly state if the chart was reproduced or simply citing

4) it is recommended to add a short subsection after 2.3 data to briefly describe how the data was input to the model, how the calculations were carried out, and the timeline of the simulation and projection.

5) Minor language check is needed, for instance, in line 40, try to avoid using "we"; in line 214, try to replace "concerning table 2" with " the key parameters are summarized in table 2"

Reviewer 3 Report

This is an excellent paper. I have two main remarks: 

1. Figures from 4 to 11 are hardly readable in black and white format. Please do add colors, instead of patterns and shading, so that each element is assigned an unique color. 

2. Discussion and conclusion should be interchanged. Place the discussion section first and add conclusion afterwards. 

Round 2

Reviewer 1 Report

I am happy to find that the authors strived to address all the issues in the manuscript.

The authors may want to carefully check the manuscript thoroughly, for instance, Line 479, "wang et al." should be Wang et al.

Best,